# A Pathway towards Climate Services for the Agricultural Sector

**Ioannis Charalampopoulos *** and **Fotoula Droulia**

Laboratory of General and Agricultural Meteorology, Department of Crop Science, Agricultural University of Athens, 11855 Athens, Greece; fdroulia@aua.gr
* Correspondence: icharalamp@aua.gr; Tel.: +30-2105294234

**Abstract:** Climate change is already having a negative impact on many areas of human activity, affecting life globally. It is more urgent than ever to increase our adaptive capacity to respond to current and future climate change risks. Climate services refer to a specialized sector that encompasses both research and operational activities. This sector is primarily focused on interpreting and communicating knowledge and information about climate risks in a manner that is tailored to meet the specific needs of diverse user communities. Climate services offer a range of specialized outputs, including forecasts, assessments, and advisories, which enable users to make decisions that are based on an understanding of the potential impacts of climate change. The outputs of climate services are designed to help diverse user communities effectively manage risks and capitalize on opportunities arising from climate variability and change. An attempt is made to outline the fundamental elements of climate services and point out their contribution to various aspects of human activity, focusing on their essential role in the adaptability of the priority for action agricultural sector, which appears as considerably vulnerable to the change of considerably susceptible to climate conditions. This article is structured to answer basic questions about climate services in general and to show the specificities of climate services in the agricultural sector.

**Keywords:** climate service; weather service; climate information; climate change; climate risk; agriculture; adaptation; agricultural adaptation; climate smart agriculture; climate services for agriculture

## 1. Introduction

A weather and climate service or Climate Service (CS) is a key tool for society to manage the risks associated with climate variability and change [1]. Nowadays, it is an important means to support individuals, communities, and organizations in effectively adapting to climate change's short- and long-term impacts (CC) [2]. The economic benefits of the CS's implementation could be vast since in 2019, only for the agriculture sector, the World Meteorological Organisation (WMO) estimated that upgraded forecasting could lead to a USD 30 billion annual increase in global productivity and a USD 2 billion reduction in annual asset losses [1]. The great importance of CS in addressing and adapting to the impacts of climate change is reflected in the studies and deliverables of major organizations such as the IPCC, FAO, UNESCO, ICOMOS, MedECC, and others [3–6].

In the past decade, there has been a notable increase in CSs research, published articles, and applications. This is due to the pressing need for quick and effective adaptation to CC. According to the Scopus citation database, an extensive number of 33,998 surveys (articles, reviews, editorials, or letters) were accessed on 9 August 2023.

Numerous complexities characterize the agricultural sector, and the end user/farmer lacks familiarity with new technologies [7,8]. In addition, agriculture constitutes a fundamental sector of human survival and the global economy since foodstuffs directly or indirectly originate from crop and animal production [9–11]. This publication attempts to provide brief answers to simple questions about climate services, specifically to those linked to the agricultural sector, such as farmers, agronomists, agriculture-related institutes,

and private companies. In addition, the article aims to provide focused information on what a CS is, what productive sectors it can help, and, most importantly, how it can provide important services to agricultural production. Ultimately, this article aspires to facilitate the reader's introduction to the complex but important world of CSs.

## 2. What Is a Climate Service?

The most common description of a CS involves the provision of climate information to assist decision-making [12]. More analytically, CSs entail generating, providing, and contextualizing information and knowledge derived from climate research aiming at decision-making across all sectors of society [13]. The CS is an information loop that links research to practical application and vice versa. Parties to this cycle can be large international organizations, countries, ministries and agencies, scientists, professionals, and even individual citizens. The parts of this cycle enable the flow of important information while at the same time enriching and modifying it to make it suitable for use in the next part of the CS. Ultimately, CS is a way and means of using climate information to make it as useful and valuable as possible.

The WMO defines CSs as "the dissemination of climate information to the public or a specific user". The CSs Partnership expands on this definition, stating that "CSs involve the production, translation, transfer, and use of climate knowledge and information in climate-informed decision making and climate-smart policy and planning" [14].

A broad meaning for the term 'CSs' is: "The transformation of climate-related data—together with other relevant information—into customized products such as projections, forecasts, information, trends, economic analyses, assessments (including technology assessments), counseling on best practices, development and evaluation of solutions and any other service in relation to climate that may be of use for the society at large" [15].

"Climate services are technology-intensive, science-based and user-tailored tools providing timely climate information to a wide set of users" [16]. Moreover, "Climate services comprise a broad collection of information service products. This diversity implies plenty of opportunities for innovations and business development. Yet the diversity in products is plagued by lack of standardization in terms of product categorization, quality assurance, etc., which seriously hampers uptake of these services" [17].

"Climate services refers to both the product, and the processes of communication and dissemination associated with a climate service. This element can be characterized by its information and knowledge (e.g., climate variables, adaptation options), format (e.g., tool, visualization), intended use (e.g., increasing awareness or supporting specific decisions), delivery method (e.g., online, workshop setting, radio) and the interactions associated with communication and dissemination (e.g., helpdesk, workshop interactions). The service product characteristics and processes may be influenced by the production processes and the use for which it is intended" [2].

"Climate services entail the transformation of climate-related data into customized products such as projections, forecasts, information, trends, economic analysis, assessments (including technology assessment), counseling on best practices, development and evaluation of solutions, and any other service in relation to climate that may be of use for the society at large" [18].

## 3. What Does a Climate Service Consist of?

A typical CS comprises a variety of tasks, information, tools, and data to offer useful insights into past, present, and future climatic conditions as well as their possible effects on different sectors and stakeholders (Figure 1). A CS aims to support decision-making, policy formulation, and adaptation strategies relating to CC and variability. Although specific services may differ, a comprehensive CS typically consists of the following fundamental elements:

One of the primary functions of CSs is data collection. Weather stations, buoys, satellites, and remote sensing technologies such as drones and airplanes equipped with

appropriate meteorological instruments are just a few of the numerous sources from which CS collect and preserve data. These data cover specifics on the following climate-related variables: air temperature, air humidity, precipitation, wind speed, and wind direction (the first left part in Figure 1). Nowadays, smartphones comprise the most essential tools for CSs' data collection and monitoring [19–21].

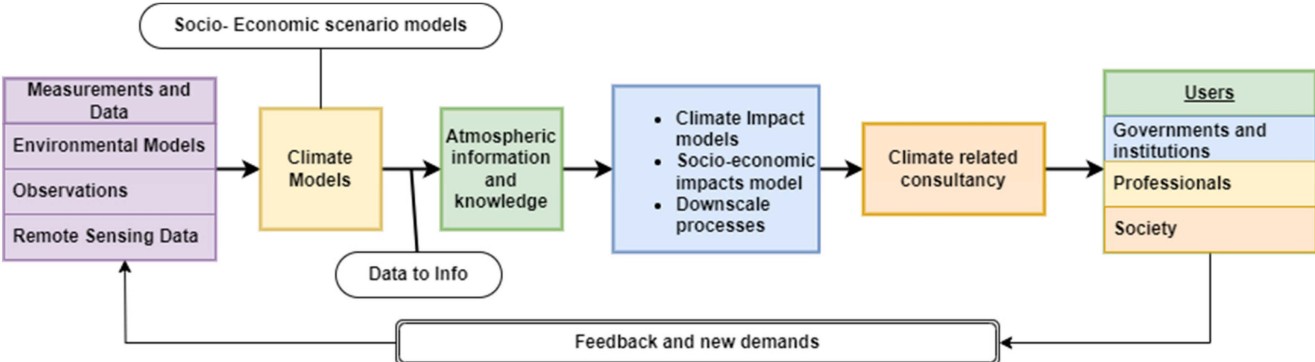

**Figure 1.** The general structure of climate services.

Moreover, CS's core is analyzing and modeling the assembled data. Climate scientists evaluate the collected data using advanced modeling approaches to comprehend past climate patterns, trends, and variability. Climate models (second part in Figure 1 in left) simulate potential future scenarios based on various greenhouse gas emission scenarios and other variables [22–24].

Another important function in the CS workflow is the tailor-made assessment, analysis, and visualization of the climatic conditions (future, present and/or past) is the third part in Figure 1. More precise, regular assessments of climate trends, impacts, and vulnerabilities are conducted to provide a clear picture of how the climate changes and how it might impact different regions, sectors, and communities [16,25,26].

In addition (the forth part in the left site of the Figure 1), a vital task of the CS processes is the risk estimation and the impact assessment on the focused sector. So, CSs evaluate the risks and effects of CC on several industries, including transportation, water management, energy, human health, and ecosystems. This information supports stakeholders in planning for adaptation and resilience measures [27–29].

One of the most important applications included in a CS is the early-warning and forecasting system, which is mainly focused on extreme weather events or adverse climatic conditions. The most frequent early-warning systems (EWS) are those that inspect heat waves, tornados, typhoons, droughts, floods, etc. The abovementioned systems are employed on a wide variety of temporal and spatial scales and can aid in effective community planning and response [30–33].

Among the paramount assistances of the CSs are the decision support tools (the last two parts of Figure 1). CS providers of CSs create approachable tools and platforms so that decision-makers in government, industry, business, Non-Governmental Organizations (NGOs), and the public can access, comprehend, and implement climate data information. These products include interactive maps, data visualization, and impact assessment tools [13,34–36].

At many levels, from local to international, CSs assist governments and stakeholders in designing policies, strategies, and plans responsive to the climate. These services aid in ensuring that accurate climatic data enable supporting options. The so-called policy and planning support of the CSs is one of the most essential tools for contemporary governance [37–40].

CSs provide training and capacity-building initiatives to enable stakeholders to understand and utilize climate data more effectively. This supports users in making defensible choices and incorporating environmental factors into their planning procedures. In this

way, a CS enhances the capacity and improves the training of the participants on climate-sensitive issues at every level [27,33,41,42].

Furthermore, critical components of an effective CS workflow are accessible communication and the extensive outreach of information and data. So, information on the climate must be communicated effectively to the public, policymakers, and stakeholders. To disseminate knowledge and advocate for climate-resilient behaviors, CSs employ a variety of communication channels, including websites, publications, seminars, and social events [43–46].

The goal of ongoing research conducted by the CSs is to enhance data collection strategies, upgrade modeling approaches, and strengthen forecasting abilities. They also look at creative methods for dealing with upcoming climate concerns. CSs nowadays are at the research and innovation tipping point, employing the state-of-the-art of digital media and worldwide big data infrastructures [15,21,23,47,48].

A CS is crucial to closing the gap between scientific understanding and real-world responses to the CC's effects. A CS is an invaluable resource for individuals, communities, corporations, and governments intending to make wise decisions in a changing world.

## 4. What Are the Main Sectors in Which Climate Services Aid?

### 4.1. Public Health

Human and livestock health is one of the most critical sectors in which CS provides essential assistance. CSs for health may be defined as "the entire iterative process of a joint collaboration between relevant multidisciplinary partners to identify, generate and build capacity to access, develop, deliver and use relevant and reliable climate knowledge to enhance health decisions" [49]. So, public health officials can forecast heatwaves, cold spells, and disease outbreaks that are impacted by climatic/weather conditions with the aid of CSs, to make decisions on diseases and illnesses related to climate variability. Effective public health initiatives and responses to safeguard vulnerable groups are made more accessible with this knowledge [50–52]. In general, CSs for public health can help to enhance human well-being and, in extreme cases, may save human lives by reducing morbidity and mortality [53].

There are several published works related to the contribution of CSs to disease limitation [50–52] and livestock disease prevention [54–56]. The CSs should be in connection with health services to assist decision-making processes and manage climate-sensitive health risks. One of the most critical targets for public health improvement in the case of CSs is the shift from health risk assessment to risk management [50].

### 4.2. Water Management

CSs are a cutting-edge tool for drinking and irrigation water management. The cardinal mission of the CS on water management is to support decision-making for the sustainable usage of this unique, essential, and vital natural resource [57,58]. Since CC drives water scarcity and/or changes in precipitable water seasonality, a CS is a way to provide accurate and informative material to users to adapt to innovative approaches in their operations. Moreover, CSs include information on reservoir levels, snowfall accumulation, and precipitation patterns. Water resource managers may use this knowledge to optimize water allocation, prepare for droughts or floods, and guarantee sustainable water utilization [58–61].

### 4.3. Disaster Management and Risk Reduction

Extreme weather events and climate-related (Figure 2) disasters pose the most significant risk to society in the last ten years [37]. Thus, having access to accurate and up-to-date climate information is essential for making better decisions about current and future weather and climate-related threats. CS for disaster risk reduction (DRR) are a nexus of actions, procedures, and connections that can organize society, institutions, and

governments to minimize the potential damage caused by adverse natural weather and climate events.

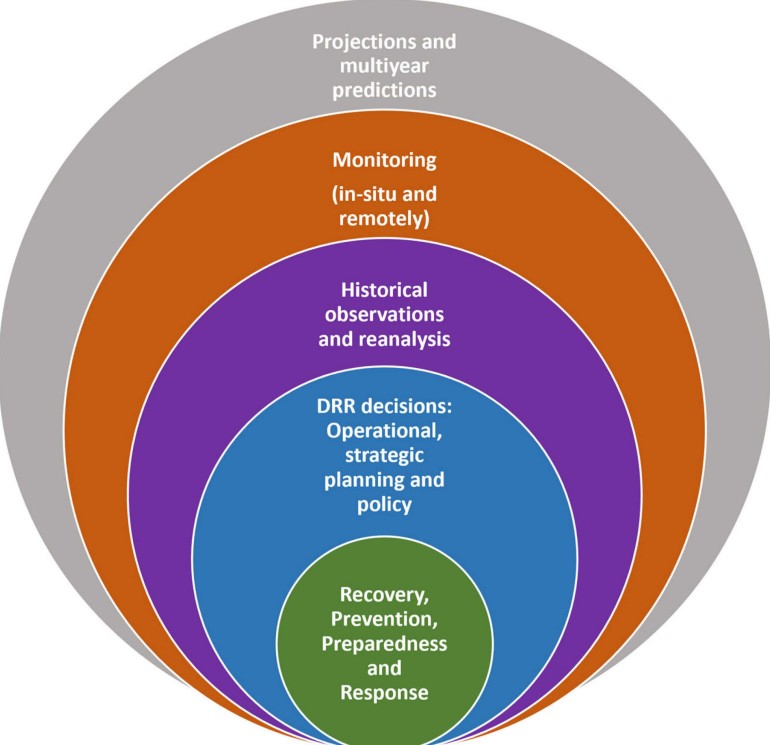

**Figure 2.** CSs' nexus for disaster risk reduction management with information from Street et al. [37].

The CC-driven disasters may befall in a wide variety of terrains, such as urban, agricultural, and natural, and may thus strike all sectors of human activities [62–64]. Generally, CSs help locate regions that are vulnerable to severe weather conditions, including hurricanes, floods, and wildfires. As such, disaster management organizations can more easily implement EWS and preparatory measures.

*4.4. Energy Sector*

Directly or indirectly, the energy sector is connected to the weather and climate. Wind, solar, and hydropower, which can be transformed into electricity, are influenced by the atmospheric conditions when the needs for buildings' heating and cooling are affected by the weather and climate [65,66]. CSs offer insights into long-term climatic patterns, assisting utilities and energy firms to plan for anticipated changes in energy demand and their effects on energy infrastructure. Since there is a high need for renewable energy (instead of energy derived from fossil fuels), weather and climate nowcasting and forecasting are of utmost importance for the energy production outlook and grid balance. At the same time, knowledge of the near future atmospheric conditions is of high value because it can help with a more accurate estimation of the energy price. The CSs are a Swiss blade tool to face the challenges of the increasing demand for buoyant renewable energy supplies [65,67].

*4.5. Urban Environment and Life*

An urban environment is highly complex and vulnerable, especially when overpopulated. Moreover, urban areas are extremely vulnerable to small-scale meteorological, hydrological, and other environmental processes [24,68]. Cities rely heavily on infrastructure, including transportation systems (road, rail, pedestrian, bicycle, etc.), water and power supply, sanitation and drainage systems, communication networks, etc. The complexity of these systems may increase the settlements' vulnerability in a nonlinear way with size; for example, doubling the size of a city may more (or less) double its complexity

and sensitivity [69–71]. The CSs, which focus on the urban climate, assist cities in facing hazards such as storm surges, floods, heat waves, and air pollution episodes, especially in changing climates. So as to achieve the previous goal, CSs should combine dense observation networks, high-resolution forecasts, and multi-hazard EWS. In order to guarantee that cities can adapt to changing climatic conditions and lower hazards to citizens, CSs assist urban planners and engineers in designing resilient infrastructure.

### 4.6. Tourism

Weather and climatic conditions exert influence on tourism on a large scale, impacting both service providers and customers. Suppliers of tourist services face a challenge because of their dual sensitivity. In addition to considering how CC and variability may drive costs, they must also consider how consumers may perceive the effects of CC on various tourist services and supporting amenities and how they might react to these effects. Although different climate-driven tourism indices have been created, consumer behavior in the industry is also influenced by a wide range of other factors, the evolution of which can be challenging to anticipate. However, given how important climate is to many forms of tourism, such as winter sports and beach vacations, the industry may greatly benefit from relevant and useful CSs, both so-called seasonal CSs and long-term CSs, for better adaptation to CC [72–75]. Finally, CSs can provide weather and climate data to the travel and tourism sector, allowing the implementation of educated decisions on travel seasons, outdoor activities, and proactive measures to avoid the visitors' endangerment associated with adverse weather [76–78].

### 4.7. Natural Areas Conservation

Forest and natural areas are of utmost importance for worldwide biology. Moreover, a sustainable and resilient economy that is climate-neutral depends heavily on the forestry industry [79]. Forests are a vital pillar for bolstering biodiversity preservation, accomplishing carbon sequestration, and enssuring the supply of diverse ecosystem services in Europe's policies by addressing environmental sustainability, biodiversity loss, and CC [80,81]. Forestry researchers consider climate data mandatory and recommend its use in forest simulation modeling and forestry impact assessment for developing climate scenarios [82]. Integrating earth observations (land cover data) and climate data has considerable potential for the forestry sector, as it incorporates CC impacts in forestry decision-making procedures and thus enhances the ecological resilience of forests (e.g., CSs providing local fire projections may assist forest fire risk management). Forestry end-users widely request the integration of climate with non-climate data. Thus, CSs suppliers should aim at evolving CSs (e.g., by creating foreseen species distribution maps, platforms on emergency risk management, and projections on bioenergy consumption) to support policy-makers in defining practice-oriented and functional adaptation strategies [83].

### 4.8. Biodiversity Conservation

Ecosystems are experiencing mounting pressure from drivers such as CC [84]. CC has directly influenced biodiversity and various ecosystem services by altering climatic elements, shifting latitudes/altitudes and phenology, and diminishing species plasticity and suitability [85,86]. As such, threats to biodiversity (CC, land use change, resource extraction, pollution, and invasive alien species) [87] should be translated into tangible and quantifiable elements that can be exploited by policy-makers for the promotion and development of flexible (adaptable), functional and effective conservation planning [88]. For instance, the Tracking Invasive Alien Species (TrIAS) project aims to inform policymakers by exploiting a data-driven workflow enabled by tracking the progression of alien species to identify the current and future risks of native species [88,89]. By employing high-resolution climate data (necessary for robust and reliable modeling of climatic suitability for invading species), estimating the potential establishment and response of specific invasive plant or animal species under CC is feasible. High-resolution risk maps corresponding to the current and

projected climate periods are exploited for expert risk evaluation of alien species based on protocols for biodiversity endangerment estimations and for conservationists' assistance in developing strategies for protecting vulnerable habitats and species [90].

### 4.9. Insurance and Finance

CCs include climate finance, since the latter involves using climate knowledge in decision-making and climate-smart policy and planning. Households susceptible to climate shocks are empowered through insurance payouts to circumvent harmful coping strategies and safeguard their resources [91]. Climate insurance, especially when weather-indexed, decodes climate information into an economic tool aiming at risk reduction and profit smoothing (e.g., household agricultural income [92]. Climate-indexed insurance addresses issues associated with traditional indemnity-based insurance, resulting in much more economical transaction costs in relation to the respective costs corresponding to traditional insurance. Private insurers are attracted by such methods while, concomitantly, clients find them more transparent than traditional insurance procedures [93].

### 4.10. International Development

CSs are critical for developing countries to access climate data, enabling them to build climate resilience, address food security, manage water resources, and plan sustainable infrastructure development [39,94]. Many developing countries are susceptible to disruptive extreme events (severe storms, droughts and floods), so the provision for early warnings is oriented towards the proactive response of government agencies, the private sector, and individuals. For instance, daily, seasonal, and annual hydrometeorology information tailored to the needs of small farmers reduces risks and increases production. Modern national hydromet systems, with the capacity to monitor, analyze, and skillfully predict weather/climate events, aid in the accomplishment of a wide range of requirements; including protecting lives (e.g., through disaster mitigation), reducing the costs of disaster relief and restoration/reconstruction on the one hand, and on the other hand, developing a large part of the population's everyday life and assisting in the nations' buoyant economy [95]. Studies on water resource management indicate the benefits of CSs associated with urban, agricultural, and environmental water use and reservoir management. Seasonal streamflow and precipitation forecasts are beneficial for informing efficient water allocation policies for agricultural production and net profit [96,97]. Many studies on the CSs' initiatives have been conducted for the agricultural sector of developing countries.

In the case of the agricultural sector, CSs can enhance the capacity to manage risks attributed to climate variability and CC by assisting adaptation decision-making and constructing resilience in the face of adversity [98]. The significance of CSs has been widely recognized, and therefore, investment in developing national meteorological services has been implemented in recent decades [99]. Agricultural information (extensively specialized climate knowledge), which is provided more effectively/operatively beyond traditional approaches, allows CSs to empower farmers by optimizing features of the crop production chain (e.g., sowing, planting, fertilizer application, irrigation, harvesting, plant protection etc.). As such, the development lies in sustainability and, most preferably, in the increase of crop production, the reduction of losses, and the decrease in input costs, thus enhancing resilience and adaptation planning [99,100].

### 4.11. Agriculture and Food Security

Given the expected major shifts in food security owing to CC, the upcoming nutrition certainty will rely on agricultural systems that are resilient to production failure when faced with crucial short-run threats [101,102]. The impacts of weather and seasonal climate extremes (drought, temperature extremes, flooding, major rainfalls and cyclones) on food consumption are widely reported, given that food security is strained in the extreme events' immediate aftermath [9,103–108]. In the concept of securing global nutrition, the association between food security and climate risk indicates that the latter's management

must be incorporated into future sustainable development goals (e.g., SDG2: "end hunger, achieve food security and improved nutrition and promote sustainable agriculture"). In this case, CSs establish their functionality and effectiveness by providing the information and support on which end users rely for comprehension, preparedness, and coping with climate-related threats in a spectrum of relevant timeframes [91].

## 5. How Do Climate Services Aid Agriculture in Adapting to Climate Change?

CSs play a vital role in helping agriculture adapt to the challenges posed by [109]. Agricultural stakeholders face increased risks and uncertainties as the climate becomes more variable and extreme weather events become more frequent and intense [110–116].

Due to its heavy dependence on favorable weather and climate parameters, agriculture may be marked as one of the main sectors that has greatly benefited from establishing CSs to some extent through the evolution of seasonal forecasts that support improved end-user decision-making [117,118]. This support is derived from the agricultural producers' improved comprehension of the strong linkage between the forecasted seasonal conditions and the impact on crop growth and development and, hence, the resolution of management practices that may potentially generate pronounced crop productivity [119,120].

CSs may assist agriculture in adapting to CC in the following domains.

### 5.1. Seasonal Climate Forecasts

CSs provide farmers with seasonal climate forecasts, indicating the expected weather patterns for a specific period, such as the upcoming growing season [121]. Services for the agricultural sector depend upon the operation of accurate procedures aiming to ensure quality seasonal forecasts. This perspective is fundamental in examined cases where, for example, the local and regional physical geography is characterized by high complexity [122]. Armed with specialized guidance on the upcoming seasonal climate conditions (e.g., knowledge of the number of rainy days and dry spells), farmers are capable of performing better adaptive farming decisions and strategies (most commonly employed: change of planting date/cultivation timing, crop acreage, land allocation, selection of the most appropriate crop type and variety, change in fertilizer quantity) and are well prepared when to expect critical weather events [123–126]. In addition, forecasts conducted by CSs have empowered developing countries to address food security through predictions of shortfalls in grain yields [127]. Globally, reliable pre-season predictions on the variations in the yields of major crops reinforce the agricultural sector's ability to respond effectively to food production shocks and to food price increases driven by extreme weather events [128].

### 5.2. Extreme Weather Event Warnings

CSs issue warnings about extreme weather events (e.g., droughts, floods, heatwaves and storms). In the context of farming systems, there has been considerable traction in evolving visualization early warning systems for climate-induced calamities across relative geographies [129]. Farmers are principally interested in shorter timescales where actions can instantly diminish the repercussions of severe weather events. As such, the exploitation of warnings by farmers and pastoralists, having improved knowledge of local climate features, may implement prompt prevention measures such as soil, water, cultivation management, and adjustment practices involving irrigation planning, sowing dates, improved varieties' selection, fertilization operations, crop diversification, herd mobility, and livestock migration (relocation of livestock to safer areas, e.g., for the avoidance of flood-prone regions) [126,130–132].

### 5.3. Crop Selection and Management

Based on climate projections, farmers can adapt their crop choices to suit the changing conditions better. Replacing a crop variety is a common tactic on the basis of the better fitting of the new variety to the projected climate (e.g., CSs can recommend drought-resistant varieties or crops with shorter growing periods to align with altered temperature

and rainfall patterns) [133,134]. Replacing a crop variety may correspond to the sowing of evolved raw material (more developed plants' seeds) or the selection of the appropriate variety mostly suited to the upcoming climatic season. The common replacement of the less climate-fitting crop variety in response to seasonal agrometeorological information may significantly increase agricultural yields [135]. Significantly increased yields of major crops (e.g., rice and maize) have already been documented and attributed to the important contribution of CSs in farming activities' planning [136], highlighting, thus, the necessity for increased access and exploitation of CSs by the agricultural community in adapting to the accelerating worldwide nutritional needs.

### 5.4. Water Management

CSs offer insights into future water availability and potential changes in precipitation patterns. Sub-seasonal-to-seasonal forecasts of daily rainfall amount and frequency, for example, could benefit rain-dependent agricultural applications [137]. This information helps farmers manage their water resources more efficiently by optimizing irrigation schedules and conserving water during dry periods based on the allowable minimization of water use (soil moisture not below a certain threshold) and thus achieving the water storage required at farm scale for its functional utilization and not at the expense of agricultural production [138].

### 5.5. Pest and Disease Management

CSs can assist in predicting changes in pest and disease patterns influenced by climate shifts [139]. Early warnings allow farmers to implement targeted pest management strategies, thus reducing the impact on crops, concomitantly preventing plant/crop disease pest outbreaks [140] and minimizing the introduction of alien plant species [141]. Early forecasting of pest and disease dynamics may promote the efficacy of prevention based on monitoring and forecasting improvements, contributing, furthermore, to the limitation of chemical pesticides' applications and consequently to food security, the sustainable development of agriculture [142], and the limitation of environmental pollution which constitutes a master anthropogenic cause of CC [143]. Climate-smart pest management (coordination of crop production, extension, investigations, and policy towards more efficient and resilient food production systems) reduces crop losses, enhances ecosystem services, minimizes the intensity of greenhouse gas emissions per unit of food produced, and reinforces the resilience of agricultural systems facing the challenge of CC [144].

### 5.6. Soil Management and Protection

Climate change is a pressing concern that affects soil health and productivity. To mitigate and adapt to its impacts, stakeholders must have access to climate services for soil protection. These services provide vital information, data, and tools that support sustainable land management practices, integrating climate data, soil information, and other relevant parameters. By integrating climate services into soil protection efforts, stakeholders can enhance the resilience of agricultural and natural ecosystems to climate change. This includes monitoring and assessing soil properties, implementing adaptive soil management strategies, and providing real-time information to respond to climate-related challenges promptly [36,145].

Moreover, stakeholders must be educated about the importance of soil protection and sustainable land management practices. Collaborating with policymakers, supporting research initiatives, and facilitating international collaboration can improve global efforts to address soil-related challenges. By prioritizing climate services for soil protection, stakeholders can ensure sustainable land use practices for the future and contribute to a healthier planet [145–148].

### 5.7. Agricultural Insurance and Risk Management

CSs support the agricultural insurance sector by providing data on climate-related risks [149]. Insurance companies can design appropriate policies that cover climate-related crop losses, offering a safety net for farmers during extreme weather events. Innovative forms of insurance based on weather parameters (weather-index insurance) protect end users/farmers against extreme weather events) [150]. Practically, such mechanisms exploit a factor (namely index as being, for example, temperature, wind speed values, or rainfall amounts) to activate indemnity payouts to farmers. In essence, determining whether farmers have suffered losses from the insured weather hazard and, therefore, accurately verifying compensation beneficiaries is possible [151]. Index-based insurance for small-holder farmers has overcome complications (e.g., moral hazard, adverse selection, increased transaction costs), which have rendered limitations to implementing traditional loss-based crop insurance. Agricultural insurance emerges, thus, as a significant user of CSs, for it complements the utilization of climate information for farm decision purposes [149]. It becomes evident that CSs support the crop insurance industry in assessing and pricing climate-related risks. In this way, insurers are assisted in offering the appropriate coverage and efficiently managing the financial risks associated with extreme weather events [152].

### 5.8. Climate-Smart Agriculture Practices

Climate-smart agriculture encourages coordinated activities by farmers, scientists, investigators, decision/policymakers, and the private/public sector towards climate-resilient routes based on four principal operation areas: "(1) building evidence; (2) increasing local institutional effectiveness; (3) fostering coherence between climate and agricultural policies; and (4) linking climate and agricultural financing". Climate-smart agriculture differs from other perspectives (e.g., the 'business-as-usual' approach) by focusing attention on the capability to apply flexible, context-specific solutions aided by contemporary policies and financing measures [153]. CSs offer guidance on climate-smart agricultural practices, which focus on sustainable and adaptive approaches to farming. This may include conservation agriculture, agroforestry, and integrated crop-livestock systems that enhance resilience to CC. In this concept, CSs comprise various environmental functions, values, and benefits, including agriculture and farming services from planting to post-harvesting activities, more sources of biomass and solar energy, ground, surface, and atmospheric water sources, product marketing, fish and livestock, water storage systems for production and private or community pastureland for grazing livestock [154]. Through climate-smart policies, the avoidance of future economic losses and the provision of the largest habitat gains for threatened species may be achieved [155,156].

### 5.9. Advisory Services

Some CSs provide personalized advisory services to farmers, tailoring climate information to their specific needs and helping them implement climate-resilient practices on their farms. Findings have revealed that favorable outcomes concerning operations to support farmers' risk decision-making and adaptation are enhanced when farmers are involved [157]. The use of information and communication technologies (e.g., cell phones, television, radio, and internet services) is an emerging field for the communication of CSs to farmers. Weather and agro-advisory services can assist rural agricultural communities in making better in-season crop management decisions, selecting technology, and developing marketing strategies [158]. On the one hand, advisory services may reduce the costs of gathering, processing, and decision-making. On the other hand, they may allow the dissemination of timely information that guides farmers in mitigating climate risks. Providing timely and relevant climate information empowers farmers to make informed decisions, minimize risks, and adapt agricultural practices to the changing climate. This, in turn, enhances the overall resilience of the agriculture sector to the challenges posed by CC. Rural advisory services (defined as: "all the different activities that provide the information and services needed and demanded by farmers and other actors in rural settings to assist them

in developing their own technical, organizational and management skills and practices so as to improve their livelihoods and well-being") [159] have a comparative advantage in technology development and information dissemination, reinforcing farmers' assistance and brokering, advocacy and policy support, ability and productivity. These goals are materialized by enhancing farming practices among individual farmers based on collaborative agricultural and institutional innovation with farmers, researchers, and other groups. Although already actively and more broadly involved in these actions, advisory services will require capacity development at individual and organizational levels and institutional adjustment at the systems level to upgrade their success and enhance innovation regarding climate-smart agriculture and CC adaptation [160–162].

*5.10. Access to Research and Technology*

CSs bridge the gap between scientific research and practical application. They make climate-related research accessible to farmers, translating scientific findings into actionable information. To address the informational challenge, the application of improved visualization techniques, easily understood by non-climate experts, has assisted end users in interpreting and exploiting knowledge as plainly and immediately as possible. Encouraging interdisciplinary groups consisting of design researchers, climate scientists, and communication specialists has generated extensive expertise and competence in all phases of CSs development [163]. Also, CSs prototypes (e.g., the Ukko project developed within the broader European project EUPORIAS), which have recognized the role of end users in co-designing the requested product, have significantly contributed to advancing the usability of climate projections, adjusting climate information to the actual demands of end users, communicating uncertainty more sufficiently and bridging the gap between state-of-the-art climate forecasts and users' preparedness for the implementation of such novel knowledge [164].

*5.11. Policy Support*

Innovative CSs, adaptation strategies, and sustained policy support are mandatory to address the challenges of establishing sustainable agriculture and food security [165]. CSs are indispensable for policymaking due to the expected increase in the variability of weather parameters, extreme climate conditions, and the vulnerability of rural communities, especially in developing countries. For instance, regional CC reflection committees have been established (e.g., in Senegal) and have requested CC be considered in local authorities' policies. The success of early weather-climate services projects has encouraged the Department of Agriculture to incorporate climate information in the National Adaptation Plan for the agricultural sector [166]. CSs projects engage policy makers to construct policies and frameworks that include CSs principles and processes involving key stakeholders' specific roles and responsibilities, which are fundamental for scaling up CS. "Scaling-up looks at challenges related to developing the institutional structures and network and enabling environment (policies, regulations, operations, and capacities). This is meant to boost the reach of the provided CS at a much larger scale (e.g., to provincial, regional, or national scale) in a sustainable way beyond piloting; it refers to the integration of the lessons learned at the local scale into the policy, practical agenda, and investment plans at higher levels" [167].

According to the abovementioned, it is evident that CSs serve as key mechanisms for the information of agricultural policies and programs, assisting governments in developing strategies for supporting climate-resilient agriculture and incentivizing the adoption of sustainable practices.

## 6. Practical Information about the Climate Services

Climate services make use of the available big data containing information on the environment and the atmosphere. In order to use this volume of data effectively, packages

and libraries have been developed in several modern programming languages, such as Python and R. This section will briefly present some of the most commonly used libraries.

**R language packages**

Clisagri: This is a package utilizing climate services focused on the agricultural sector, containing functions for the modeling of climate-related risk and quantification of drought, wetness, heat stress, and cold stress exposure. Also, Clisagri offers dynamic phenological modeling and application for optimal variety selection and crop cycle duration.

According to the developers, Clisagri offers the agriculture industry user-friendly, co-designed climate service products. In order to evaluate risks and crop exposure, it provides a simple method for obtaining important agro-climate indicators and makes dynamic phenological models available. It is possible to use and test a wide variety of crop varieties; in fact, the present version of Clisagri has this capability for wheat. Additionally, it enables users to quickly analyze how crop development is affected by past, present, and future climatic circumstances. Although several pre-processing procedures (such as skill assessment and bias adjustment) are not yet accessible, climate projections and predictions can still be utilized [168].

Climate: This is a package focused on accessing free meteorological and hydrological datasets in a fast and consistent way. The core of the Climate package functionality is the automation for downloading global meteorological data such as Ogimet, vertical soundings measurements from Wyoming University, and more.

According to the package developers, users can get historical and current meteorological data from the ground and higher atmosphere using the climate R package. Climate data downloads allow for the intuitive and fully automated application of atmospheric data gathered in compliance with WMO standards. The package is intended for environmental scientists who want to collect meteorological or hydrological data for research reasons easily and it can be easily programmed using the R programming language. It is designed to be user-friendly. For many non-atmospheric scientists who struggle with the usually complex and time-consuming procedures for accessing in-situ atmospheric data in a ready-to-use structure, the usefulness and simplicity of the suggested approach can be quite valuable. Time can be saved by using the climate package's suggested solution [169].

CSTools: The CS Toolbox is designed to assess and improve the quality of climate forecasts for seasonal and multi-annual scales. Along with basic and sophisticated techniques to obtain personalized results, the package includes process-based, state-of-the-art methods for forecast calibration, bias correction, statistical and stochastic downscaling, optimal forecast combination, and multivariate verification. The package developers conclude that CSTools includes state-of-the-art methods for post-processing seasonal forecast datasets, emphasizing statistical correction, downscaling, and classification methods [170].

CSIndicators: This R package compiles generalized techniques for the adaptable calculation of indicators linked to climate change. Additionally, every embodied technique, which is a distinct mathematical methodology, is complemented by the option to choose an adaptable time frame for defining the indicator. The above makes it possible to customize indicators for sector-specific climate service applications in various ways. Although the algorithms in this package are designed for sub-seasonal, seasonal, and decadal climate predictions, they can also be used for other time scales. Furthermore, this tool works well with the CSTools R package for post-processing climate forecasts. Finally, the developers of the CSIndicators mentioned that the package is compatible with other tools related to wind energy applications and, of course, the agricultural sector [171].

The Climate Data Tool (CDT): This R package provides an easy-to-use and freely available tool for performing all the necessary actions to collect, organize, and analyze climatic and meteorological data for climate services. The CDT has already been used by 24 African countries. Along with the core functions, CDT provides a graphical user interface (GUI) for use by stakeholders without coding skills. Moreover, CDT is capable of providing tools for assessing and downloading raw data, conducting quality control, checking the accuracy of coordinates, and making basic statistical analyses. Moreover, the user can

perform gridding analysis, spatial interpretation, and merging station observations and proxy data along with data validation [26].

**Python libraries**

XCast: According to the developers, this library simplifies the climate forecasting process. It consists of a user-friendly API that leverages Xarray and Scikit-Learn to make it easy to use. It's also compatible with Dask, allowing easy scaling to institutional supercomputers. XCast has lowered barriers to computational earth science by simplifying workflows. This library may be a core tool for implementing operational climate services [172].

xclim: This xarray-based climate data analytics library allows for the computation of climate indicators over large, heterogeneous data sets. It is built using xarray objects and operations, which enable seamless parallelization handling provided by dask (a Python library for parallel computing). Additionally, xclim relies on community conventions for data formatting and metadata attributes. The purpose of xclim is to serve as a tool for facilitating climate science research and delivering operational climate services and products. Besides climate indicator calculations, xclim includes utilities for bias correction and statistical adjustment, ensemble analytics, model diagnostics, data quality assurance, and metadata management [173].

C3S: The Copernicus Climate Change Service, called C3S, is a Python based toolbox that provides access to open and free state-of-the-art climate data and tools for the use of a wide range of stakeholders (researchers, scientists, private entities, public authorities, and governments). The European Union funds the European Centre, which implements this service for Medium-Range Weather Forecasts (ECMWF). The C3S was established in 2014, and the Climate Data Store became available in 2018. The last is the core of the C3S, a cloud-based infrastructure providing access to a vast range of global and regional information related to weather and climate. The system is planned and implemented to be accessible to nonspecialists using Python scripts [174].

IMPACT2C: Last but not least, there is the IMPACT2C web-atlas—Conception, which is a web-based climate service product focused on the interdisciplinary and harmonized format. The available content of this climate service is focused on supporting decision-makers in policy negotiations and raising awareness about climate change issues. Technically, the IMPACT2C combines content management and geographic information system, including distinguished sectors for tourism, energy, health, agriculture, forest and ecosystems, water, and coastal areas. The whole service can be utilized with Python scripts to derive tabular and map data for stakeholders [175].

The following Table 1 contains some of the biggest and most well-organized generic climate services and some of the most well-known climate services dedicated to the agricultural sector.

**Table 1.** A sample of the most well-known climate services.

| CS Name | Short Description | Link |
|---|---|---|
| Copernicus (Climate Change) Services | The Copernicus infrastructure for climatic data dissemination. | https://www.copernicus.eu/en/copernicus-services/climate-change (accessed on 23 December 2023) |
| European Flood Awareness System | The web-based system for preparatory measures before major flood events. | https://www.efas.eu/en (accessed on 23 December 2023 (accessed on 23 December 2023) |
| The European Forest Fire Information System | A digital toolbox for the monitoring of forest fires. | https://effis.jrc.ec.europa.eu/ (accessed on 23 December 2023) |
| The European Drought Observatory | The web-based drought-relevant information infrastructure. | https://edo.jrc.ec.europa.eu/edov2/ (accessed on 23 December 2023) |
| My Climate View | A climate digital toolbox for the farmers of Australia. | https://myclimateview.com.au/ (accessed on 23 December 2023) |

**Table 1.** *Cont.*

| CS Name | Short Description | Link |
|---|---|---|
| BlightSpy | A climate service for the prediction of blight. | https://blightspy.huttonltd.com/ (accessed on 23 December 2023) |
| CLIMALERT | Web-based climate services focused on sustainable water and agriculture. | https://jpi-climate.eu/project/climalert/ (accessed on 23 December 2023) |
| Agrometeorological Indicators Explorer and Data Extractor | A repository of the agrometeorological data in a worldwide spatial scale. | https://cds.climate.copernicus.eu/cdsapp#!/software/app-agriculture-agera5-explorer-data-extractor?tab=app (accessed on 23 December 2023) |
| U.S. Climate Resilience Toolkit | A web-based climate toolbox that helps people to build climate resilience in one easy-to-use location. | https://toolkit.climate.gov/ (accessed on 23 December 2023) |
| U.S Drought Monitor | Digital map for drought monitoring on a weekly basis. | https://droughtmonitor.unl.edu/ (accessed on 23 December 2023) |
| Climate in a Glance | A web page containing climatic data presented in maps, timeseries and tabular formats. | https://www.ncei.noaa.gov/access/monitoring/climate-at-a-glance/ (accessed on 23 December 2023) |

## 7. Discussion and Conclusions

Principal impediments to the functional and equitable communication of CSs may be practically summarized as the lack of interaction between CSs' producers and end users, lack of awareness about the existence of particular climate knowledge, the absence of national capacity for communication, lack of user-orientation of services, deficient interpretation of services into actionable products, lack of understanding and capacity to use climate information, poor comprehension of scientific uncertainties, and the strong digital divide across and within countries (e.g., owing to policies or regulations that inhibit the introduction of digital technology) [3,27]. The absence of internet access in the least developed countries' rural areas raises considerable concern since the greater connectivity advantages are demonstrated for remote agricultural areas composed of populations with low computer literacy and limited formal education [176].

In the agriculture sector, CSs frequently fail to make it to the 'last mile' (a term comprehended as small-scale agricultural producers often living in remote areas beyond the reach of public services). Thus, the co-design and co-production of CSs constitute further constraints, particularly regarding incorporating climate information into planning and decision-making procedures that underpin all the farm-level critical decisions [3].

Since the turn of the century, researchers, technical staff, scientists, donors, and NGOs have significantly contributed to reinforcing the development of both physical and technical capacities to produce the most accurate climate information. However, improvement of coordination in the group of diverse actors engaged with CSs and communicating climate knowledge to smallholder farmers and pastoralists remain crucial challenges, given that much more progress is required, especially in developing countries [135,177,178]. For instance, further research on modalities for investigating options for using innovations (e.g., mobile applications used to disseminate agricultural information to smallholder farmers) to upgrade the cooperation between CS suppliers and receivers is becoming more essential [179]. The effectiveness of the achievement of, e.g., mobile phones utilized in the communication of weather information to smallholder farmers (through variants, e.g., as Short Message Services/SMS, calls, voice mail, or the internet), depends upon the socio-economic settings of the farmers, including the language and climate information's accuracy, the farmers' literacy level and participation in the CSs' co-designing, confidence in the provider and response-ability [180].

Information and communication technologies (ICTs) are characterized by vast potential in enabling real-time communication of climate information and agricultural counseling to farmers. However, access to ICTs in the least developed countries is constrained by the communities' poverty and the disproportional exclusion of rural women and youth.

Restricted access to information in rural communities is attributed to the elevated costs and absence of infrastructure, extending from discontinuous electricity supply to highly reduced availability of ICT facilities [176]. Improvements in "agricultural production, input supply, agricultural research and national agricultural information systems, extension and advisory services, postharvest processes, weather information gathering and dissemination, and agricultural disaster management" comprise potential advantages of the digital technology implementation referred to by the FAO e-agriculture strategy [181]. In addition, the overcoming of the challenge for the CSs' availability and accessibility by the most vulnerable groups may seriously contribute to the achievement of various sustainable development goals (SDG) (e.g., specifically of the no poverty-SDG 1, zero hunger-SDG 2 and climate action-SDG 13) [3].

Furthermore, the exploitation of multidisciplinary platforms for facilitating and ensuring institutional coordination and project sustainability can be prosperous, principally where institutional disconnects and human capital limitations obstruct the successful application of agrometeorological services. Substantial prospects for modifying productivity and livelihoods are demonstrated by the enhanced services provided. These services overcome complexities involving data quality, availability, and access and concomitantly promote stakeholder engagement and use, as well as the characterization of climate risks at a local scale encouraging further applications and research trends [182,183].

The integration of instant crisis responses with distant future adaptation features corresponding to achieving a more holistic climate risk approach necessitates proper enabling parameters. For instance, a supportive and encouraging policy-making environment, sufficient funding, political participation, commitment and design, decision-makers' scientific and technical capacity, active support of key stakeholders, and climate alertness constitute CSs' more comprehensive future approach [184,185].

Aiming at boosting the CSs' effectiveness in the agricultural area, service providers should host a variety of experts (e.g., social managers, environmental engineers, communication consultants, and analysts) along with stakeholders representing the corporate world and public services. International collaboration must be considered a prerequisite for reinforcing local and regional capacities in developing countries [44] where the agricultural sector constitutes the primary source of livelihoods [186].

In addition, the heterogeneity between end users and distinct requirements among individuals should be considered, given that a broader inclusion is crucial to circumventing the reinforcing of power imbalances [187]. In the agriculture sector of developing countries, for instance, where gender determines the tasks undertaken by men and women, gender differences are reflected in information needs and differing abilities to act on that information (e.g., gender-sensitive CSs for women who demonstrate different weather and climate information requirements) [188]. Also, it is underlined that the needs and interests of end users are not static (changing expectations also of funders and providers) but may be diversified in the course of time by the potential overcoming, e.g., of social, educational-literacy level and economic constraints [189].

It is recognized that, although major gaps in their development remain on a global level, CSs have demonstrable benefits to agriculture and food security by navigating agricultural producers around unforeseeable and dynamic climate/weather patterns. Investments in CSs that concentrate on the "last mile" can seriously contribute to constructing buoyant and sustainable food systems [3]. Finally, new future challenges are expected to arise in the concept of the necessity for common standards for CSs (e.g., ethics, language, and terminology) and for consistent and coherent services across weather and climate time periods [39].

The effective handling of these challenges aimed at national agricultural development presupposes the commitment of the community of scientists and practitioners to engage in reflective and crucial debates (reinforced by policies' sensitivity to climate information and assistance) based on the future conceptualization and operationalization of agricultural CSs.

In general, CS is a combination of new technologies and methods that aim to make better use of the vast amount and high quality of climate and weather information that is currently open and available. The great advantage of CS is its adaptability to the needs of the industry you want to serve. The use of new technologies, such as smartphones and the internet, has made it possible to create and operate CS, and AI will make them even more effective. All productive sectors can benefit from CS because they all depend directly or indirectly on weather and climate. The agricultural sector is obviously benefiting immensely from Cs because it is the sector that is highly dependent on weather and climate, and producers have not had the tools and skills to make use of the relevant data. Investing in CS related to the agricultural sector and food production can have multiplier economic benefits in terms of ensuring production and global food sufficiency. It is obvious that CS will grow quickly and efficiently because the technologies that support them are becoming more and more economical while at the same time they can be used by non-experts. Additionally, climate change mitigation and adaptation need tools and methods that work continuously and provide a continuous and two-way flow of information for all stakeholders.

**Author Contributions:** Conceptualization, I.C.; methodology, I.C. and F.D.; writing—original draft preparation, I.C. and F.D.; writing—review and editing, I.C. and F.D.; visualization, I.C.; supervision, I.C. All authors have read and agreed to the published version of the manuscript.

**Funding:** This research received no external funding.

**Data Availability Statement:** Data availability upon request.

**Conflicts of Interest:** The authors declare no conflict of interest.

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
