# Peer review of "A Pathway towards Climate Services for the Agricultural Sector"

_climate, doi:10.3390/cli12020018_

Round 1

Reviewer 1 Report

Comments and Suggestions for Authors

The paper needs concrete examples or cases to show the significance and easy-operation of  the service.

1. What is the main question addressed by the research? it is more about a way for agricultural sector to apply climate services,which is an interesting issue.

2. Do you consider the topic original or relevant in the field? Does it

address a specific gap in the field? YES

3. What does it add to the subject area compared with other published

material? The realted study has less attention to this issue.

4. What specific improvements should the authors consider regarding the

methodology? What further controls should be considered? Concrete applied examples for CS should be listed.

5. Are the conclusions consistent with the evidence and arguments presented and do they address the main question posed?

Must be improved for easy application.

6. Are the references appropriate? should be updated. Comments on the Quality of English Language

Moderate editing is necessary.

Reviewer 2 Report

Comments and Suggestions for Authors

Please see the enclosed form

Reviewer 3 Report

Comments and Suggestions for Authors

Dear authors,

In this manuscript, you present in a comprehensive way the possibilities of these tools, which would significantly help agricultural producers in overcoming the challenges they face due to unexpected climatic conditions. In this connection, and what you have shown in the manuscript, more attention should be paid to the following questions:

Sustainable resource management and environmental protection for agricultural production is the basis for ensuring long-term food security.

Protection and improvement of soil quality: improving soil health, diversity and fertility, optimizing the use of nutrients in the soil, increasing the content of organic matter, reducing the likelihood of disease, pests and weeds and at the same time reducing the use of plant protection products.

Protection and improvement of the quality and structure of the soil, increasing the water capacity of the soil, protecting the soil from erosion and loss of nutrients and controlling weeds without the use of chemicals.

Improving biodiversity on agricultural land, through the creation and maintenance of new habitats for pollinators, other beneficial insects and birds, which contributes to increasing their number and diversity, then preventing soil erosion and improving the rural environment.

Preservation and improvement of overall biodiversity in meadows and pastures through low-intensity grazing throughout the year and limitation of meadow mowing during the breeding season of wild animals and field birds. Also, the goal of this operation is to create optimal conditions for the reproduction of predatory insects that are useful on agricultural land and to protect the natural floral composition of permanent grasslands, which contributes to maintaining meadows and pastures in good ecological condition and preserving the characteristics of rural landscapes.

Chapter 4 is not necessary, it should focus exclusively on agriculture in accordance with the title of the work, although agriculture is closely related to other branches of the economy. My suggestion is that you remove this chapter from the manuscript, and that in another chapter, if possible, point out the possible benefits in these domains that are listed above.

Reviewer 4 Report

Comments and Suggestions for Authors

The manuscript covers a highly critical issue on contribution of Climate Services to agricultural sector. Some changes are required for the manuscript to be further improved:

1. Please extend the Abstract section since it has to be at about 200 words.

2. Please extend the Introduction section to be sure that the most updated improvements on the study field are included.

3. All lines of the manuscript that do not include text (blank lines), such as #56, 57, 59, 88, 89, 90 etc. should be removed.

4. Line#58: The title could be removed since the whole Section 2 provides definitions of Climate Services.  

5. Figure 1 should be removed from Section 2 into Section 3 which is more relevant.

6. The different elements of Climate Services presented in Section 3 should be connected to each different element shown in Figure 1.

7. In Section 4, there is no need to present all sectors affected by Climate Services since the manuscript presents "A pathway towards climate services for the agricultural sector" (title of manuscript).

7.1. Section 4.1 "Public health" could be removed since it is not related to the manuscript main target.

7.2. Section 4.3 should be solely focused on impacts of climate extremes on agriculture; otherwise, it has to be removed.

7.3. Section 4.4 "Energy sector" should be solely focused on agriculture.

7.4. Section 4.5 "Urban environment and life" could be removed since it is not related to the manuscript main target.

7.5. Section 4.6 "Tourism" could be removed since it is not related to the manuscript main target.

7.6. Section 4.7 "Natural areas conservation" could be removed since it is not related to the manuscript main target.

7.7. Section 4.8 "Biodiversity conservation" should be solely focused on agriculture.

7.8. Section 4.9 "Insurance and Finance" should be solely focused on agriculture.

7.9. Section 4.10 "Internation development" should be solely focused on agriculture.

8. Lines #321-326. You include information on "energy" in the section of "inetrnational development". Please remove the lines.

9. Lines #327-330. You include information on "tourism" in the section of "inetrnational development". Please remove the lines.

10. Line #332. You mention Figure 3 without including the figure within the manuscript.

11. Lines #335-345. Please remove this paragraph into Section 4.11 "Agriculture and food security".

12. Line #363. Please mention or even describe the challenges.

13. Line #364. Farmers are part of a whole agricultural stakeholders group. Please rephrase.

14. Line #368. Please place reference #120 in the correct location within the sentence.

15. Lines #571-572. Please rephrase the title.

16. Lines #615-620 mention that some specific use cases are presented in the paper without however, providind more details. Please remove the lines.

17. Table 1 constitutes the core part of the manuscript. Please create a much more detailed table, providing additional information on the specific data provided by each Climate Service, time scales, availability (free or paid). By this way, the impact of the manuscript to the scientific community would be surely increased.

18. Line #691. Please rephrase the title into "Discussion and Conclusions".
